# Unilateral Osteonecrosis of the Femoral Head in a Patient with Atopic Dermatitis Due to Uncontrolled Topical Steroid Treatment, a Case Report

**DOI:** 10.3390/reports8020065

**Published:** 2025-05-11

**Authors:** David Glavaš Weinberger, Lena Kotrulja, Snježana Ramić, Patricija Sesar, Slaven Babić

**Affiliations:** 1Traumatology Department, Sestre Milosrdnice University Hospital Center, 10000 Zagreb, Croatia; slaven.babic@kbcsm.hr; 2DermaPlus Clinic, 10000 Zagreb, Croatia; lena.kotrulja@gmail.com; 3Pathology and Cytology Department “Ljudevit Jurak”, Sestre Milosrdnice University Hospital Center, 10000 Zagreb, Croatia; snjezana.ramic@gmail.com (S.R.); patricijasesar@yahoo.com (P.S.); 4Department of Biology, Faculty of Science, University of Zagreb, 10000 Zagreb, Croatia; 5School of Medicine, Catholic University of Croatia, 10000 Zagreb, Croatia

**Keywords:** femoral head osteonecrosis, atopic dermatitis, topical corticosteroids, adverse effects, steroid-induced avascular osteonecrosis

## Abstract

**Background and clinical significance:** Osteonecrosis of the femoral head (ONFH) is a disease of the epiphysis caused by the death of osteocytes and osteoblasts, resulting in debilitating pain. ONFH can be traumatic or nontraumatic, with prolonged glucocorticoid use being the leading cause of nontraumatic ONFH. Atopic dermatitis (AD) is a chronic inflammatory skin condition typically treated with topical corticosteroids. ONFH following topical corticosteroid treatment is exceedingly rare, with limited documentation in the literature. We present a case of an under-recognized complication of prolonged topical corticosteroid treatment. **Case presentation:** We report a case of a 29-year-old Caucasian male patient with sharp right hip pain. Plain radiographs, a CT scan, and an MRI indicated Ficat and Arlet stage 3 ONFH. The patient reported the prolonged uncontrolled use of topical mometasone furoate for five years due to AD. Following the diagnosis, topical corticosteroids were discontinued, and the treatment was shifted to tacrolimus and, subsequently, to oral methotrexate with folic acid. The patient underwent a total hip arthroplasty in June 2022. Given his young age and poor response to previous treatments, he was transitioned to upadacitinib, which led to significant improvement without skin flare-ups or postoperative hip pain. **Conclusions:** This case highlights the rare, but serious, risk of ONFH associated with long-term topical corticosteroid use. It underscores the importance of monitoring systemic side effects in dermatological therapies and educating patients on proper corticosteroid use. Alternative treatments, such as upadacitinib, should be considered in young male patients to prevent severe complications.

## 1. Introduction and Clinical Significance

Osteonecrosis of the femoral head (ONFH), also known as avascular necrosis, is a progressive disease characterized by the death of subchondral osteocytes and osteoblasts due to a disrupted blood supply [1]. This disruption leads to subchondral fractures, bone collapse, and debilitating pain. ONFH can be classified into traumatic and nontraumatic causes, with prolonged glucocorticoid use being the most common cause of the nontraumatic form [2]. The Ficat and Arlet classification is widely used to describe the stages of ONFH, considering features observed on plain radiographs, MRI, and clinical indicators such as pain, limping, and stiffness [3,4].

Atopic dermatitis (AD) is a chronic, inflammatory skin disease that commonly starts in early childhood, but can persist throughout life and is characterized by recurring episodes of flare-ups [5]. The course of the disease is relapsing, and it is frequently associated with elevated levels of serum immunoglobulin E (IgE) and an individual or family history of type I allergies, allergic rhinitis, or asthma [6]. Atopic dermatitis severity is commonly measured using standardized scoring systems, such as the Eczema Area and Severity Index (EASI) and the SCORing Atopic Dermatitis (SCORAD) index which assess the extent and severity of skin involvement [7,8].

The long-term treatment of AD with topical corticosteroids may lead to several iatrogenic conditions, such as epidermal thinning, Cushing’s syndrome, and diabetes [9,10].

ONFH following topical corticosteroid therapy has rarely been reported in the literature, and while systemic glucocorticoids are a well-known cause of ONFH, the role of topical corticosteroids in this condition remains poorly understood. Given their widespread use, particularly in AD, it is essential to assess their potential systemic effects.

We report a case of a 29-year-old man with AD who developed ONFH following the long-term application of topical steroids due to flare-ups of the disease.

## 2. Case Presentation

A 29-year-old Caucasian male patient presented to the emergency room in March 2022 with progressively increasing pain in his right hip over two months. Upon physical examination, the patient was limping and had extensively painful and limited rotations of the right hip joint. He was referred for an X-ray, a computed tomography (CT) scan, and magnetic resonance imaging (MRI) of the pelvis and both hips. The changes were indicative of stage 3 (Ficat and Arlet classification) ONFH of the right hip (Figure 1). The left femoral head was unaffected.

### 2.1. History and Symptoms

The patient admitted having a lifelong history of AD since toddlerhood and reported exacerbations of inflamed skin lesions over the past three years, despite not undergoing a dermatological checkup since 2017. He disclosed a daily use of steroid creams for the last three years, comprising approximately 3 to 5 g per day (an estimated 90 to 150 g per month) of a moderate-potency topical corticosteroid (mometasone furoate 0.1% cream) to the affected areas, primarily to the face, upper back, antecubital, popliteal fossa, armpits, and groin. He was otherwise healthy and conducted no risky behavior.

Due to his history of AD and uncontrolled use of corticosteroid creams, he was referred to the dermatology department.

### 2.2. Skin Examination and Laboratory Results

Upon dermatological examination, the patient presented with diffuse skin dryness and inflamed erythematous papules scattered over his face, arms, legs, and trunk, along with pruritic lichenified plaques in the antecubital and popliteal fossae. The prolonged use of steroid cream resulted in skin atrophy and the appearance of distended striae in the skin flexure areas (Figure 2). The patient also reported being previously diagnosed with allergic rhinitis.

We conducted an initial clinical assessment using the EASI and the SCORAD index, which revealed a moderate disease severity with an EASI score of 26.7 and a SCORAD index of 33.3. These findings, along with a review of the patient’s medical history, confirmed the diagnosis of AD.

Laboratory tests revealed peripheral blood eosinophilia (eosinophils: 10.3%, range: 0.0–6.0%), an elevated total serum immunoglobulin E (IgE) (level of 3188 kIU/L, range: <160 kIU/L), and elevated specific IgE against various airborne allergens (ambrosia, grass pollen, birch pollen).

Upon the diagnosis of ONFH, the patient discontinued using steroid creams and was transitioned to non-steroidal moisturizers and a topical calcineurin inhibitor ointment (0.1% tacrolimus) for the management of AD. The patient received counseling on proper skincare practices and the potential long-term consequences of topical steroid overuse. The crescent sign was present at the anterolateral aspect of the right femoral head on plain radiographs, indicating Ficat and Arlet stage 3 ONFH. The crescent sign was a linear cleft due to a subchondral bone fracture, which may lead to the collapse of the femoral head. A total hip arthroplasty (THA) was indicated as a curative procedure for this patient’s hip illness, given the presence of femoral head collapse. He underwent a THA in June 2022. A direct lateral (Hardinge) approach was used. The acetabular component (Continuum, size 52, Zimmer Biomet, Warsaw, IN, USA) was placed using the line-to-line technique, while an uncemented Avenir, with a size 5 femoral stem (Zimmer Biomet, Warsaw, IN, USA) and a 36+4-sized head, was placed in the femoral canal (Figure 3).

### 2.3. Imaging and Pathological Findings

The femoral head was sent for a pathohistological analysis, which showed necrosis of the trabecular bone, dilated and empty lacunae with pyknotic nuclei (osteocyte apoptosis), and fat infiltration of the bone marrow (Figure 4 and Figure 5). These findings ultimately confirmed the radiological diagnosis of ONFH.

### 2.4. Treatment Course

During the hospital stay, the patient received multimodal opioid-sparing analgesia according to the “fast track” protocol and was allowed to walk on the day of surgery using crutches and weight bearing as tolerated. After fulfilling the criteria, the patient was discharged. For the next two weeks, celecoxib 200 mg once daily and acetaminophen 1000 mg were prescribed for analgesia. The patient continued physical therapy at home.

Due to persistent severe itching and the insufficient effect of the topical calcineurin inhibitor treatment on the inflamed skin lesions, the patient started immunosuppressive therapy with methotrexate (MTX). Oral MTX and folic acid once-a-week therapy were gradually introduced from October 2022, with an initial dose of 7.5 mg of MTX, increased to 15 mg, and 5 mg of folic acid. However, after three months of the MTX treatment, the skin lesions persisted, along with intense itching, resulting in a clinical EASI score of 19.9 and a SCORAD score of 29.2. Due to the poor effectiveness of conventional immunosuppressive therapy with MTX and considering the patient’s reproductive age, the Janus kinase 1 (JAK1) inhibitor upadacitinib was introduced at a dose of 30 mg daily in February 2023. Regular dermatological follow-ups were conducted every 4 months. Upon the patient’s last orthopedic follow-up visit in November 2024, the patient was completely pain-free and with no limitations regarding their range of motion and everyday activity.

Significant improvement was observed at the patient’s dermatological exam in February 2024, with an EASI score of 0.6 (Table 1). The patient reported no itching or adverse effects since starting the new therapy regimen with the JAK1 inhibitor upadacitinib.

## 3. Discussion

Systemic corticosteroid treatment is reported to be the leading cause of iatrogenic ONFH; however, it is exceedingly rare for it to occur after topical corticosteroid treatment. Cumulative dosing and the long-term use of topical corticosteroids present a significant risk in the pathogenesis of ONFH [11].

Between 20,000 and 30,000 cases of ONFH are detected each year in the United States alone, accounting for 8.3% of all THAs [2]. Nontraumatic ONFH generally affects a younger and more active population between the ages of 20 and 40, impacting the contralateral femoral head in 70% of cases [12]. A “multi-hit” hypothesis is applicable to the pathogenesis of ONFH, since there are often concomitant factors, such as genetic and epigenetic mutations and comorbidities, that lead to the development of osteonecrosis [13]. There are several therapeutic options available. Milder cases (Ficat and Arlet stages I and IIa) can be treated with core decompression. A THA is the therapeutic option of choice when a crescent sign is present or there is further collapse of the femoral head [14]. Generally, ONFH is described as steroid-induced when patients have been on a glucocorticoid therapy of >2 g prednisolone (or its equivalent) daily within a 3-month period, if the diagnosis of ONFH has been made within <2 years from the initiation of glucocorticoid therapy, and if the patients do not have additional risk factors [15].

ONFH after topical corticosteroid treatment has been reported in cases where corticosteroids have been used to treat various skin conditions.

Cunliffe et al. [16], Hogan et al. [17], Tang et al. [18], and Takahashi et al. [19] have reported five patients and seven hips affected by ONFH due to topical corticosteroid application for the treatment of psoriasis. McLean et al. [20] and Kubo et al. [11] reported two patients and three hips affected by ONFH after receiving corticosteroid ointment treatment for eczema. In all cases, ultra-high- and high-potency topical corticosteroids were used.

Hogan et al. reported the case of an ethanol abuser using 250 g per month of 0.05% clobetasol propionate ointment over a period of 5 years, resulting in a cumulative dose of 15,000 g of clobetasol propionate [17]. The patient reported by Tang et al. used 2.5 g daily of 0.025% fluocinolone on his hands over the course of four years, amounting to a total of 3560 g [18]. Kubo et al. reported a case of a patient with bilateral osteonecrosis of the femoral head following the use of 2–3 g per day of 0.05% clobetasol propionate on his face and neck over a period of 2 years and 10 months, resulting in a cumulative dose of 2530 g of clobetasol propionate. They also reported no other risk factors for the patient [11]. Our patient had no other risk factors and reported using 3 to 5 g of 0,1% mometasone furoate daily for the last 3 years, which amounts to a cumulative dose of 4380 g over the course of three years. Given the use of topical corticosteroids since early childhood, measuring the cumulative dose over his lifetime is challenging due to changes in the total body surface area over time.

A recent study on the Korean population found that systemic corticosteroid use (≥1800 mg) was significantly associated with the development of ONFH, while cumulative doses of corticosteroids did not influence the progression to a THA [21].

The prevalence of AD in early childhood is estimated at 15% to 30%, while in adults, it ranges between 1% and 3% of the population. About 20% to 30% of childhood cases persist into adulthood [9,22]. The AD pathophysiology is complex, resulting from skin barrier dysfunction and an unregulated immune response. Genetic and environmental factors influence this condition [23].

Topical corticosteroids are the mainstay in AD treatment. They may also carry the risk of systemic absorption, primarily when used for a prolonged duration or applied to thin and sensitive skin areas such as the eyelids, face, and flexures [24].

The percutaneous absorption of topical corticosteroids may vary based on factors such as the potency, formulation, site of application, and skin condition. The impaired barrier function of the skin significantly enhances steroid absorption, leading to increased systemic exposure [25]. Skin absorption through AD skin is deemed to be nearly doubled compared to healthy skin, both intra- and extralesionally [26]. This absorption poses risks of both cutaneous and systemic adverse events, including the suppression of the hypothalamic–pituitary–adrenal axis and related complications, particularly with the use of potent or superpotent topical corticosteroids over an extended period [9]. Chronic exposure to corticosteroids, even cutaneous, may induce a microvascular compromise through several mechanisms: fat embolism, bone marrow fat cell hypertrophy, elevated intraosseous pressure, the inhibition of angiogenesis, and the glucocorticoid-induced apoptosis of osteocytes. These factors collectively impair bone perfusion and remodeling, contributing to the pathogenesis of osteonecrosis [27].

Tacrolimus is a non-steroidal topical calcineurin inhibitor that acts as an immunomodulator and effectively addresses the signs and symptoms of AD, reduces the frequency of flares, and provides the possibility of achieving long-term disease control [9]. Low-dose methotrexate is an additional therapeutic option for AD. It is an immunosuppressive drug that works by suppressing T-cell activation [28]. However, its long-term use is limited due to potential adverse effects, including hepatotoxicity and myelosuppression [29]. Young male patients should be carefully counseled during their methotrexate therapy course due to the potentially irreversible impact on spermatozoal DNA [30]. Upadacitinib is a selective JAK1 inhibitor that works by inhibiting key cytokines involved in AD, such as IL-4 and IL-13 [31]. It is effective for treating moderate to severe AD in patients who have not shown an adequate response or intolerance to the first- or second-line agents [32]. This targeted approach provides faster relief from pruritus and inflammation, improving the quality of life more rapidly compared to conventional immunosuppressive treatment [31].

The EASI and the SCORAD index are both validated tools used to assess the severity of atopic dermatitis and monitor the treatment response [7,8]. The EASI focuses on objective clinical signs: redness, thickness, scratching, and lichenification across four body regions [7]. The SCORAD index combines both objective signs (i.e., the extent and intensity of lesions) and subjective symptoms (i.e., itchiness and sleep disturbances) [8]. The observed reduction in the EASI and SCORAD scores in our patient indicated a significant clinical improvement after upadacitinib therapy, reflecting a decreased disease severity and an enhanced quality of life.

## 4. Conclusions

This work presents a rare case of ONFH associated with the use of moderate-potency topical steroids for AD. Although rarely, continuous topical corticosteroid therapy may induce similar pathological conditions, such as those seen with systemic administration. The cause of ONFH as a systemic side effect could be due to a compromised skin barrier function in those with AD, leading to the increased percutaneous absorption of prolonged topical steroid treatment. This case highlights the importance of regularly monitoring patients receiving long-term corticosteroid therapy for dermatological conditions to minimize the risk of complications, regardless of their potency. Early recognition and intervention are crucial for minimizing morbidity, preventing contralateral ONFH, and improving the outcomes in affected individuals. Clinicians should be vigilant for signs of steroid-induced adverse effects and educate patients on proper application and the potential risks associated with the prolonged use of topical corticosteroid treatment.

## Figures and Tables

**Figure 1 reports-08-00065-f001:**
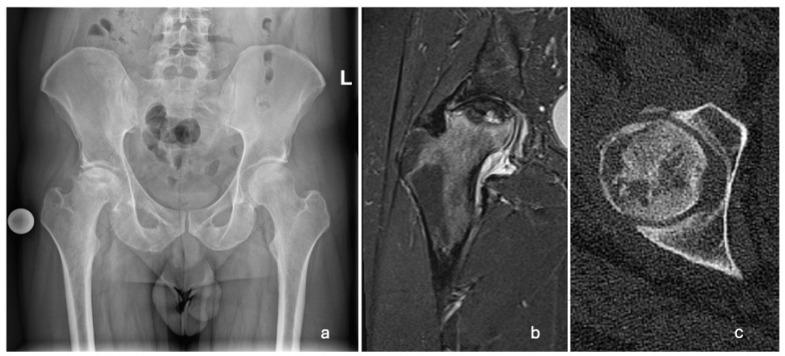
Radiographic images showing unilateral ONFH of the right femoral head. (**a**) AP pelvis X-ray taken in March 2022, showing a Ficat and Arlet stage 3 right ONFH; (**b**) T2-weighted MRI of the right hip taken in April 2022, showing ONFH with a subchondral fracture and femoral head collapse >2 mm (Association Research Circulation Osseous stage 3B); (**c**) CT scan of the right hip showing cystic deformation, the loss of continuity, and the applanation of the anterolateral femoral head.

**Figure 2 reports-08-00065-f002:**
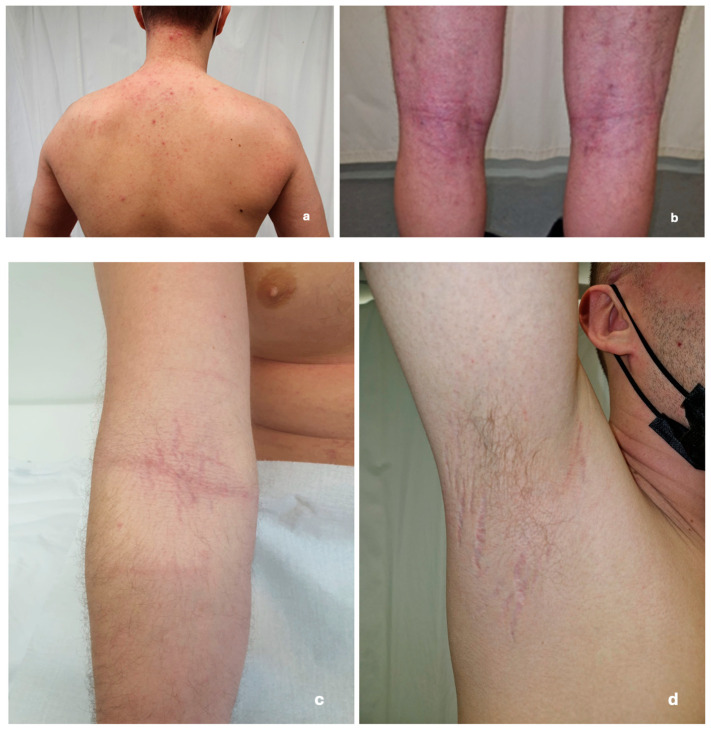
Images of skin changes taken during the dermatological exam in March 2022. (**a**) Inflamed erythematous papules scattered over the skin of the back; (**b**) lichenified erythematous plaques on the popliteal fossa; (**c**) distended stretch marks in the cubital fossa (excessive use of topical steroids); and (**d**) distended stretch marks in the armpit.

**Figure 3 reports-08-00065-f003:**
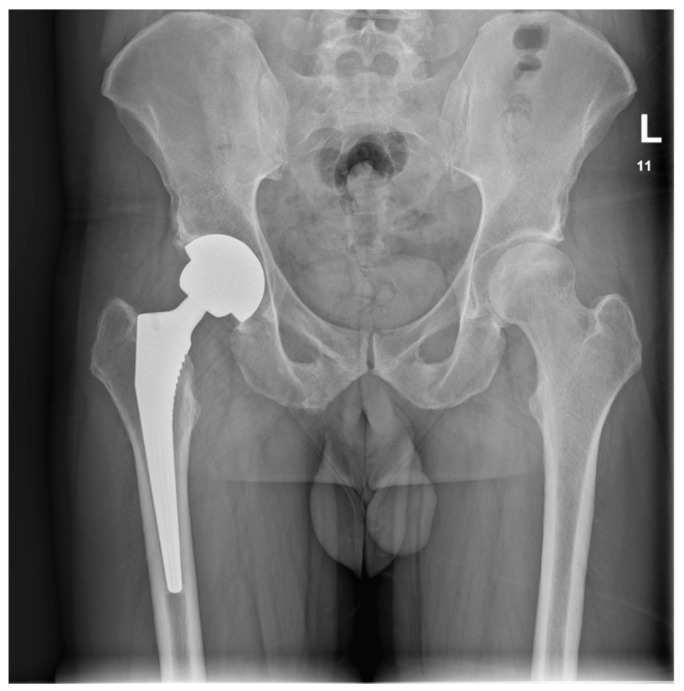
Postoperative AP pelvis X-ray taken in June 2023, showing a stable THA.

**Figure 4 reports-08-00065-f004:**
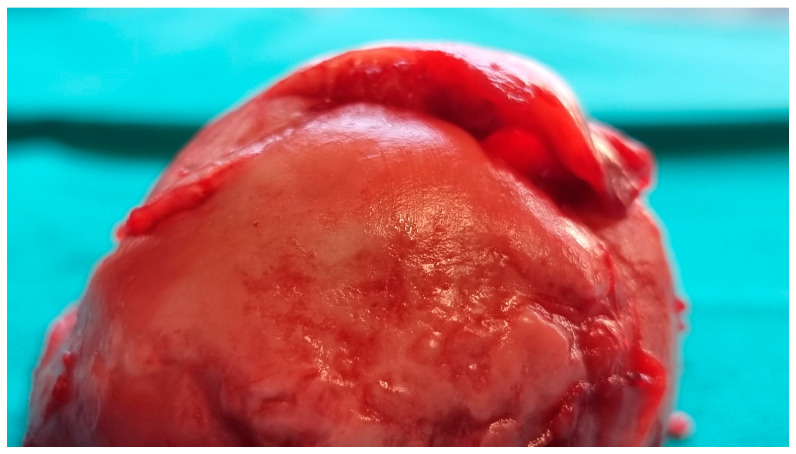
Intraoperative image of the resected right femoral head showing subchondral bone collapse and cartilage detachment.

**Figure 5 reports-08-00065-f005:**
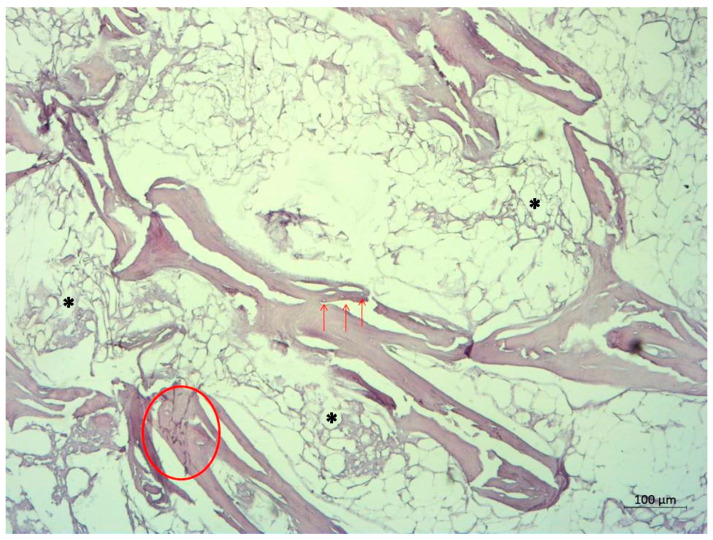
Pathohistological image of the femoral head, displaying basophilic trabecular bone (circle) representing necrosis. Arrows indicate empty and dilated lacunae, while asterisks indicate fat tissue necrosis. The absence of blood vessels in the bone marrow is notable.

**Table 1 reports-08-00065-t001:** EASI and SCORAD scores of the patient throughout the treatment.

Score (Range)	June 2022	October 2022	February 2023	February 2024
EASI (0–72)	26.7	22.8	19.9	0.6
SCORAD (0–103)	33.3	45.3	29.2	N/A

N/A—not available.

## Data Availability

The original data presented in the study are included in the article, further inquiries can be directed to the corresponding author.

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
