# Peer review of "Unilateral Osteonecrosis of the Femoral Head in a Patient with Atopic Dermatitis Due to Uncontrolled Topical Steroid Treatment, a Case Report"

_reports, 2025, doi:10.3390/reports8020065_

Round 1
Reviewer 1 Report
Comments and Suggestions for Authors
Dear Authors,
Thank you for sharing your case report. The topic is interesting, and the case provides valuable insights into the potential systemic effects of long-term topical corticosteroid use. Below are some comments and suggestions to improve the clarity, structure, and completeness of the manuscript:
Abstract
- Line 16: Please revise "FHON" to "ONFH," as "Osteonecrosis of the Femoral Head (ONFH)" is the standard abbreviation in academic and clinical literature. This change will improve clarity and consistency with existing research.
- Line 25: Specify the classification system used for stage 3 ONFH. It is likely the Ficat and Arlet classification; please explicitly state this.
- Line 29. Remove " (THA)." Since total hip arthroplasty (THA) is only mentioned once in the abstract
Introduction
- Line 42: Use "Osteonecrosis of the femoral head (ONFH)" instead of "femoral head osteonecrosis (FHON)." ONFH is the standard term and will improve readability and keyword search efficiency, as mentioned in the abstract section
- Line 53: Spell out the full names of EASI (Eczema Area and Severity Index) and SCORAD (SCORing Atopic Dermatitis) upon their first appearance.
- Line 69: Spell out the full names of CT (computed tomography) and MRI (magnetic resonance imaging) upon their first appearance.
- Line 76: Spell out the full name of ARCO (Association Research Circulation Osseous) upon its first appearance.
- Line 90: Correct "present" to "presented."
- Line 80 ~112: This paragraph is lengthy and could benefit from being divided into smaller, more focused sections for improved readability. Suggested sections include: History and Symptoms, Skin Examination and Laboratory Results, Imaging and Pathological Findings, and Treatment Course.
- Line 112-113: Briefly describe the Ficat and Arlet classification and explain its diagnostic significance in this case.
- Line 113: Explicitly list all medical devices used, including brand names where applicable.
- Line 113. Specify the surgical approach used for the total hip arthroplasty (e.g., posterolateral, anterolateral, direct lateral, direct anterior, transtrochanteric).
Discussion
- Line 165: Given that approximately 70% of ONFH cases involve both hips, please describe the evaluation of the patient's contralateral hip.
- Line 172: Regarding reference 13, the cited statement '[a glucocorticoid therapy of >2g prednisolone (or its equivalent) daily for at least 3 months]' is inaccurate. The reference specifies a cumulative dose of >2g prednisolone (or its equivalent) within a 3-month period, not 'at least' 3 months.
- Line 177 ~180: The literature review mentions studies reporting ONFH after topical corticosteroid use. Please provide references for any observational studies that found no significant association. In addition, I suggest that the authors enhance the discussion by comparing the characteristics of their patient with those described in other case reports. This comparison should specifically highlight the variations in topical steroid usage, including brand of topical steroid, dosage, duration, application area, or other concomitant medication.
- Expand on the rationale and effects of immunosuppressive therapy and JAK1 inhibitors in this case.
- When discussing changes in EASI and SCORAD scores, briefly explain how these indices reflect the patient's clinical improvement.
Discuss the pathophysiological mechanisms by which ultra-high and high-potency topical corticosteroids may contribute to the development of ONFH.
Author Response
Dear editor, thank you for the positive review and for the insights on what to additionally analyze in our study. We believe you provided valuable feedback, and we have responded accordingly. Please see the attachment for detailed responses to your comments.

Reviewer 2 Report
Comments and Suggestions for Authors
Summary of the Manuscript
This case report presents a rare instance of unilateral femoral head osteonecrosis (FHON) in a young adult patient with atopic dermatitis (AD), attributed to prolonged and unsupervised use of topical corticosteroids. The authors provide a comprehensive account of the clinical presentation, diagnostic evaluation, and therapeutic management, including the transition from corticosteroid therapy to systemic immunomodulatory treatment and eventual total hip arthroplasty. The manuscript addresses a clinically significant and underreported complication, highlighting the systemic risks associated with topical therapies and underscoring the critical importance of patient education.
Strengths of the Manuscript
Originality and Clinical Relevance:
The manuscript addresses a rare but important adverse effect of chronic topical corticosteroid use, a subject with limited prior documentation in the literature.
It provides interdisciplinary insight, bridging the fields of dermatology and orthopedics.
Clarity and Structure:
The manuscript is well-structured and clearly articulated.
The case presentation is logically sequenced and supported by appropriate diagnostic imaging and therapeutic rationale.
Educational Value:
The report emphasizes the importance of educating patients on the appropriate use of topical corticosteroids.
It raises awareness among clinicians regarding potential systemic complications arising from treatments typically considered localized.
Use of Evidence and Literature:
The discussion is well-grounded in current literature and effectively contextualizes the case.
The authors appropriately acknowledge the rarity of this complication and refrain from overgeneralization.
Recommendations for Revision
Clarification of Corticosteroid Dosage and Frequency:
This submission lacked specific details regarding the dosage and frequency of mometasone furoate application. We suggest that the authors address this gap by providing essential context for evaluating the potential for systemic absorption.
Expanded Discussion on Pathophysiology:
We recommended elaborating on the mechanisms by which topical corticosteroids may lead to systemic complications such as osteonecrosis. We suggest that the authors incorporate current literature on percutaneous absorption and steroid-induced microvascular compromise.
Enhanced Imaging Description:
The radiological findings were improved by specifying disease staging according to the Ficat classification and referencing distinct imaging characteristics that supported the diagnosis.
Final Recommendation:
Accept with Minor Revisions
The manuscript fulfills the journal’s standards for originality, clinical significance, and scholarly rigor. Following the implementation of minor revisions, the case report is deemed suitable for publication.
Author Response
Dear editor, Thank you for the positive review and for the insights on what to analyze further in our study. We believe you provided valuable feedback, and we have responded accordingly. Please see the attachment for detailed responses to your comments.

Round 2
Reviewer 1 Report
Comments and Suggestions for Authors
Thank you to the authors for addressing the previous comments. The manuscript has been significantly improved and is now suitable for acceptance. However, a few minor issues remain that should be corrected:
1. Line 51: In the sentence: “The Ficat and Arlet classification is widely used to describe stages of ONFH, considering features observed on plain radiographs, MRI, and clinical indicators such as pain, limping, and stiffness [3, 4]”, please add a period at the end.
2. In line 141:“and fat infiltration of the bone marrow. (Figures 4 and 5).” Please remove the redundant period after “marrow“.
3. Lines 204–205: Please delete “Fare clic o toccare qui per immettere il testo..” and place the period correctly after the citation: “…[16].”
Author Response
Dear editor, thank you again for reviewing our manuscript and suggesting the necessary corrections. The responses to your comments are as follows:
Comment 1: Line 51: In the sentence: “The Ficat and Arlet classification is widely used to describe stages of ONFH, considering features observed on plain radiographs, MRI, and clinical indicators such as pain, limping, and stiffness [3, 4]”, please add a period at the end.
Response 1: As you suggested, we have corrected this and added a period at the end of the sentence. The change is in line 50.
Comment 2. In line 141: “and fat infiltration of the bone marrow. (Figures 4 and 5).” Please remove the redundant period after “marrow“.
Response 2: Thank you for noticing this. We have deleted the redundant period after "marrow". You can see the change in line 144
Comment 3. Lines 204–205: Please delete “Fare clic o toccare qui per immettere il testo..” and place the period correctly after the citation: “…[16].”
Response 3: Thank you for pointing this out. An automatic message from the reference manager was probably overlooked during the corrections. We have worked to correct this, as shown in lines 204-205.
We have also conducted English language corrections using the Authors' service provided by MDPI. All the corrections are still visible in track changes.